# The Impact of Surgical Techniques in Patients with Rectal Cancer on Spine Mobility and Abdominal Muscle Strength—A Prospective Study

**DOI:** 10.3390/cancers14174148

**Published:** 2022-08-27

**Authors:** Iwona Głowacka-Mrotek, Michał Jankowski, Bartosz Skonieczny, Magdalena Tarkowska, Tomasz Nowikiewicz, Łukasz Leksowski, Mariusz Dubiel, Wojciech Zegarski, Magdalena Mackiewicz-Milewska

**Affiliations:** 1Department of Rehabilitation, Nicolaus Copernicus University in Toruń, Collegium Medicum in Bydgoszcz, 85-094 Bydgoszcz, Poland; 2Department of Surgical Oncology, Nicolaus Copernicus University in Toruń, Collegium Medicum in Bydgoszcz, 85-094 Bydgoszcz, Poland; 3Department of Urology, Nicolaus Copernicus University in Toruń, Collegium Medicum in Bydgoszcz, 85-094 Bydgoszcz, Poland; 4Department of Gynecology and Oncological Gynecology, Nicolaus Copernicus University in Toruń, Collegium Medicum in Bydgoszcz, 85-094 Bydgoszcz, Poland

**Keywords:** spinal mobility, abdominal muscle strength, rectal cancer, abdominoperineal resection, anterior resection

## Abstract

**Simple Summary:**

The aim of our study was to evaluate the spine joint mobility, muscle strength and chest mobility in patients undergoing surgery for colorectal cancer following abdominoperineal resection, or laparoscopic or open resection of the rectum. The studied patients were examined three times: prior to surgery, three months after surgery and six months after surgery. Three months after surgery, all study groups showed a reduction in the range of spine joint mobility, reduction in chest mobility and a reduction in the rectus abdominis and oblique muscle strength. Six months after surgery, an improvement was observed in terms of mobility and muscle strength. The dynamics of improvement were the greatest in patients undergoing laparoscopic anterior resection.

**Abstract:**

The aim of this non-randomized study was to evaluate the impact of spine joint mobility and chest mobility on inhalation and exhalation, and to assess the abdominal muscle strength in patients undergoing surgery for colorectal cancer with one of the following methods: anterior resection, laparoscopic anterior resection or abdominoperineal resection. In patients who were successively admitted to the Department of Surgical Oncology at the Oncology Center in Bydgoszcz, the impact of spine joint mobility, muscle strength and chest mobility on inhalation and exhalation wasassessed three times, i.e., at their admission and three and six months after surgery. The analysis included 72 patients (18 undergoing abdominoperineal resection, the APR group; 23 undergoing laparoscopic anterior resection, the LAR group; and 31 undergoing anterior resection, the AR group). The study groups did not differ in terms of age, weight, height, BMIor hospitalization time (*p* > 0.05). Three months after surgery, reductions in spine joint mobility regarding flexion, extension and lateral flexion, as well asreductions in the strength of the rectus abdominis and oblique muscles, were noted in all study groups (*p* < 0.05). In comparison between the groups, the lowest values suggesting the greatest reduction in the range of mobility were recorded in the APR group. Surgical treatment and postoperative management in colorectal cancer patients caused a reduction in spine mobility, abdominal muscle strength and chest mobility. The patients who experienced those changes most rapidly and intensively werethose undergoing abdominoperineal resection.

## 1. Introduction

Colorectal cancer is the third most common type of cancer in the world [1]. Improvements of treatment outcomes, 5-year survival and quality of life in patients undergoing surgery for colorectal cancer pose a challenge for surgical oncology.

Modern radical treatment of colorectal cancer is based on surgical resection. The most important factors determining the choice of the surgical technique include staging, tumor location, grading, anal sphincter function and the patient’s general condition. Surgical resection for rectal cancer may also be performed incompletely in palliative care, and the intervention is part of symptomatic treatment. The most frequently performed interventions include abdominoperineal resection, anterior resection and Hartmann’s procedure. All those procedures can be performed with the traditional open approach or laparoscopy. It is commonly accepted that laparoscopic surgical procedures have benefits such as: less blood loss during surgery, less pain postoperatively, faster recovery of intestinal function after surgery [2,3] and patients undergoing this type of surgery have similar long-term overall survival and recurrence [4] and a better quality of life [5].

In therapy, apart from surgical procedures, radiotherapy and chemotherapy are also applied in many patients. They are used additionally before surgery as part of neoadjuvant treatment or in palliative care, regardless of the choice of surgical technique (laparoscopy or open surgery) [6]. Surgery, chemotherapy andradiotherapy all can have a negative effect on respiratory, circulatory and musculoskeletal systems [7]. During surgery, the skin, subcutaneous tissue, fascia, rectus muscles and the oblique and transverse abdominal muscles are transected. Surgery and neoadjuvant treatment also have a negative impact on physical activity and make the patient weaker [8]. Patients after surgery also limit their physical activity, which increases the postoperative complication rate and diminishes muscle mass and strength [9]. An earlier study comparing mobility in the spine joints and muscle strength in patients undergoing surgery for colorectal cancer with either laparoscopic or open rectal resection showed better results in patients treated with laparoscopy in the early postoperative period [10]. The earlier study did not provide information on how the parameters of spine mobility, abdominal muscle strength and chest mobility change in the long term. The assessment of muscle strength and mobility of the spine has important clinical significance. Studies show that decreased spinal mobility and decreased abdominal muscle strength are factors affecting postural changes [11], spinal deformities [12], degenerative disc disease as well as gait disorders and an increased risk of disability [13]. Moreover, adequate mobility of the spine indirectly impacts quality of life [14]. Following the ERAS (enhanced recovery after surgery) protocol is a very important aspect of patient preparation prior to surgery. The elements of the ERAS protocol include increased physical activity, improved nutrition, cessation of smoking and abstinence from alcohol, preoperative equalization of anemia, prophylactic antibiotic therapy, carbohydrate drinks being administered before the procedure, avoidance of premedication, postoperative prevention of nausea and vomiting and early mobilization of the patient after the procedure. Several benefits, including faster functional recovery, have been demonstrated in clinical trials and meta-analyses in patients in whom the ERAS protocol had been followed [15,16].

Considering the above-listed facts, it is important to assess patient mobility, muscle strength and respiratory function depending on the type of surgery performed. The aim of the study was to assess the mobility of the spine joints, the mobility of the chest forinhalation and exhalation and the strength of the abdominal muscles in patients undergoing surgery for colorectal cancer using the following surgical techniques: anterior resection of the rectum, laparoscopic anterior resection and abdominoperineal resection.

## 2. Materials and Methods

This prospective study (per-protocol analysis) was approved by the Bioethics Committee at the Nicolaus Copernicus University in Toruń (Decision no. 283/2019). Informed consent was obtained from each participant. The enrollment of patients started on 1 June 2019 and ended on 31 March 2020. The recruitment of patients was originally planned to be carried out until 31 May 2020, but due to the coronavirus pandemic, it was terminated earlier. Consecutive patients admitted to the Department of Surgical Oncology of the Oncology Center in Bydgoszcz were enrolled in the study. Those patients were qualified for standard resection procedures (anterior resection, laparoscopic anterior resection and abdominoperineal resection). The analysis of the range of mobility in the spine joints andthe assessment onthe mobility of the chest and the strength of the rectus and oblique muscles were performed on the day of admission to the department to determine the baseline condition, as well as three months and six months after surgery. Even though the ERAS protocol had not been followed in full, a number of its elements, including the avoidance of preoperative fasting, restrictive fluid therapy, prompt restitution of oral nutrition, shorter periods of abdominal drain and intravesical catheter use, avoidance of gastric probes being used as a standard measure and timely discharge, had been used in standard management. Figure 1 shows the recruitment of patients to the study in detail.

Before surgery, patients underwent diagnostic work-up to determine the stage of the disease and their general health. It included a colonoscopy with histopathology confirmation, imaging studies (pelvic MRI or CT, thoracic and abdominal CT), ECG and blood tests. All patients were consulted by the anesthesiologist to determine the risk of perioperative complications using the ASA (American Society of Anesthesiologists) classification. The following clinical information was also used for the statistical analysis: hospitalization time after surgery, the presence of complications after surgery, the staging of cancer and the type of pre- and postoperative treatment.Following the diagnostic tests, all patients were qualified for potential neoadjuvant treatment administered when a stage II or III tumor (cTNM: cT3–4, N0, M0 or CT1–4, N+, M0) had been confirmed. Neoadjuvant treatment included irradiation according to a 5×5 Gy protocol with or without sequential chemotherapy (3 courses of 5-FU with leucovorin) or radiochemotherapy (50.5 Gy in 28 fractions of 1.8–2 Gy with simultaneous 5-FU-based chemotherapy). Follow-up examinations were performed 11 weeks after the initiation of radiotherapy, and a surgical procedure was performed in the absence of complete clinical response 12 weeks after the initiation of radiotherapy.

The following inclusion and exclusion criteria were used in the study protocol:

### 2.1. Inclusion Criteria

-consent for participation;-age over 18;-colorectal cancer patients;-patients qualified for the following procedures: abdominoperineal resection, anterior resection and laparoscopic anterior resection;-primary surgery for colorectal cancer;-mobile patients with a good performance status (ZUBROD score 0);-preoperative clinical stage I–III;

### 2.2. Exclusion Criteria

-metastatic cancer—CS IV;-ASA 4 or more;-psychiatric disorders;-malnourishment (defined by ESPEN). The following malnourishment criteria were assumed:∗worsening nutritional status and BMI (body mass index) < 18.5 kg/m^2^;∗unintentional weight loss of >10% regardless of time or >5% over the past 3 months;∗BMI < 20 kg/m^2^ in patients <70 years of age, or <22 kg/m^2^ in people over 70;-patients who were reoperated on after primary surgical resection;-patients who required conversion during the surgical operation.-During surgery, the following techniques were applied:-anterior resection (AR)—The cut ran along the midline below the umbilicus over 15 to 25 cm. During the procedure, the following parts of the rectum were removed: 1–5 cm below the tumor up to the rectosigmoid junction together with the distal sigmoid and mesorectum;-laparoscopic anterior resection (LAR)—The technique involvedthree small cuts (approx. 1.5 cm long) in the right lower quadrant, another cut in the left lower quadrant 8–10 cm long and an oblique incision on the left side of the rectus abdominis muscle;-abdominoperineal resection (APR)—The cut was made in the midline below the umbilicus over 15 to 25 cm. As a result, an ostomy to the sigmoid in the left lower quadrant was created. The second cut was made in the perineum in the location of the excised anus over 10 cm. During the procedure, the following parts were removed: the rectum with the mesorectum and anus up to the rectosigmoid junction together with the distal sigmoid.In all patients with APR, the perineal part was performed in the gynecological (lithotomy) position, whereas the abdominal part was performed in the supine position.

The examination protocol was designed as follows. Before the first evaluation, the patient’s height and weight were measured, and BMI was calculated. Then, the measurements were taken using a tape measure. Those included the mobility of the spine joints, mobility of the chest and abdominal rectus and oblique muscle strength, and they were taken three times: prior to surgery, three months after surgery and six months after surgery.

-thoracic spine flexion:the initial measurement was from Th1 (spinous process of the first thoracic vertebra) to Th12 (spinous process of the 12th thoracic vertebra), and the final measurement was from Th1 to Th12 in a forward bending position. The mobility of the thoracic spine wasthe difference between the final and the initial measurement.-lumbar spine flexion:the initial measurement was from L1 (spinous process of the first lumbar vertebra) to L5 (spinous process of the 5th thoracic vertebra), and the final measurement was from L1 to L5 in a forward-bending position. The mobility of the lumbar spine wasthe difference between the final and the initial measurement.-total spine flexion: the distance between the external occipital protuberance to the median sacral crest. Then, the same distance was measured after asking the patient to fully bend over. The total spine mobility wasthe difference between the final and the initial measurement.-lumbar spine extension: the distance between the tip of the xiphoid process and the pubis. Then, the same distance was measured after asking the patient to bend backward. The difference between the final and the initial measurement was defined as the lumbar spine extension range.-lateral flexion of the thoracolumbar spine: the distance between the armpit and the iliac crest was measured while standing. Then, the patient was asked to bend to the side, and the distance was measured again. The mobility was defined as the difference between the final and initial measurements. The measurements were taken on both sides.-rotation of the thoracolumbar spine: the distance between the tip of the xiphoid process and the anterior superior iliac spine was measured and was then again measured after the patient rotated the torso in the opposite direction. The distance was measured on the right and left side. The mobility was defined as the difference between the final and initial measurements.-The chest circumference was measured, and then the chest circumference was measured again forinhalation and exhalation. The difference between inhalation and exhalation was calculated and defined as chest mobility.-the strength of the rectus abdominis muscle:the evaluation was made in the supine position with the lower limbs bent at the knee and hip joints, and then the patient was instructed to bend forward. At the same time, the apparatus was placed on the rectus abdominis muscle.-the strength of the oblique abdominal muscles was assessed in the supine position. The patient had lower limbs bent at the knee and hip joints, and then they were ordered to bend over to the right and left knee. While bending, the apparatus was placed on the right and then on the left oblique muscle, and the results were written down.

Rectus and oblique muscle strength was assessed using the Micro Feet Muscle Strength Apparatus, a handheld muscle strength tester.

### 2.3. Statistical Analysis

The statistical analysis was performed using the PQStat statistical package version 1.8.2.232.The differences between the groups of patients in terms of age, weight, height and BMI were analyzed by a one-way analysis of variance and a posthoc Tukey test, and in the case of hospitalization time, the analysis was performed using aKruskal–Wallis test and a posthoc Dunn test with Bonferroni correction.

For qualitative variables such as sex, pTNM, pre- and postoperative treatment, complications and disease progression, the analysis was performed using achi-squared test or Fisher’s exact test.

Joint mobility and muscle strength results were compared between the patient groups using the Kruskal–Wallis test, and the differences between measurement dates were analyzed by the Friedman test using the Dunn test with Bonferroni correction as a posthoc analysis for both comparisons.

The probability of *p* < 0.05 was considered significant, and the probability of *p* < 0.01 was considered highly significant.

## 3. Results

The analysis included 72 patients (18 undergoing abdominoperineal resection, the APR group; 23 undergoing laparoscopic anterior resection, the LAR group; and 31 undergoing anterior resection, the AR group).

The study group was analyzed in terms of clinical data. The groups did not differ with respect to age, weight, height, BMI and hospitalization time (*p* > 0.05). The results are shown in Table 1.

There were no statistically significant differences between sexes (*p* = 0.4935), the type of postoperative treatment (*p* = 0.5161) and the presence of postoperative complications (*p* = 0.2607). Postoperative complications were classified according to a modified Clavien–Dindo scale. Grade II–IV complications included 3× surgical site infections, 2× postoperative obstructions and 1× stoma necrosis in the APR group; 3× intestinal anastomosis leak in the LAR group; and 4× surgical site infections, 3× intestinal anastomosis leaks, 1× drain displacement and 1× pelvic abscess in the AR group. Statistically significant differences between the groups were noted for preoperative treatment (*p* = 0.0439) and cancer staging (*p* = 0.0466). The results are shown in Table 2.

The same letter at two median values indicates no significant difference (*p* > 0.05); if no same letters are used with two median values, the difference between the compared medians (distributions) is statistically significant (*p* < 0.05).

Table 3 presents the results of measurements of spine mobility and of the assessment of abdominal muscle strength and chest mobility. Before surgery, no statistically significant differences were found between the study groups in any of the studied parameters (*p* > 0.05). On the second and third examination, statistically significant differences were noted for all tested parameters except for the difference between inhalation and exhalation. The lowest mobility of the spine on the second and third examination was noted in the group after abdominoperineal resection for all the studied parameters compared to the baseline condition.

Three months after surgery in the APR group, there was a statistically significant reduction inall tested parameters. Six months after surgery, the mobility results decreased significantly for the following parameters: total spine flexion, thoracic spine flexion and lumbar spine flexion. For lumbar spine extension and left thoracic spine rotation, the average value stayed constant, andfor the thoracolumbar spine lateral flexion, thoracolumbar spine lateral flexion, thoracolumbar spine rotation, inhalation–exhalation difference, rectus abdominis muscle strength and right and left abdominal oblique muscle strength, the results improved in a statistically significant way.

In the LAR group, statistically significant differences were noted three months after surgery for all parameters, except for thoracic spine flexion (*p* < 0.05). For the second examination, there was a decrease in the results for the studied parameters. Six months after surgery, statistically significant differences were also noted for all parameters; in comparison to the second examination, an improvement in spine mobility can be seen for the following measurements: total spine flexion, thoracic spine flexion, lumbar spine extension, right thoracolumbar spine lateral flexion, left thoracolumbar spine lateral flexion, right thoracolumbar spine rotation, left thoracolumbar spine rotation, rectus abdominis muscle strength, right and left abdominal oblique muscle strength, inhalation–exhalation difference as well as increases in abdominal muscle strength and chest mobility. The mean result for the lumbar spine flexion remained the same.

Three months after surgery in the AR group, there was a statistically significant deterioration in the results for all measured parameters, which indicates a reduction in mobility in the spine joints, andsix months after surgery, there was a statistically significant improvement for the following parameters: total spine flexion, lumbar spine extension, inhalation–exhalation difference, rectus abdominis muscle strength and right and left abdominal oblique muscle strength. For the rest of the parameters, the results were the same as they were three months after surgery. The results are shown in Table 3.

We analyzed the results of total spine flexion, inhalation–exhalation difference and rectus abdominis muscle strength in terms of neoadjuvant or adjuvant treatment. No statistical differences were found. The results are presented in Table 4.

## 4. Discussion

We analyzed patients who subsequently qualified for rectal resection procedures (LAR, AR and APR), usually performed as part of radical treatment for rectal cancer. The patients were assessed before surgery, three months after surgery and six months after surgery. The studied groups of patients did not differ in terms of clinical parameters such as: age, weight, height and BMI, anddifferences in cancer staging based on the postoperative histopathological study (ypTNM, pTNM) did not affect the extent of surgical treatment. In the AR and LAR groups, similar rates were found for preoperative radiotherapy, which in turn was used much more commonly in the APR group, which may suggest a more aggressive therapy.

Our own study confirmed the reduction in the muscular strength of rectus and oblique muscles after surgery in patients undergoing surgery for rectal cancer, regardless of the type of surgery performed. Muscle strength was the most diminished in patients treated with APR. Six months after surgery, the muscle strength improved in all studied groups. However, the greatest improvement was noted in the group of patients undergoing laparoscopic anterior resection, but even then, the condition did not return to the values before the procedure in any group. In our study, a reduction in chest mobility (difference between inhalation and exhalation) was also noted. In their study, Sanver et al. showed that expiratory muscle strength and maximal oxygen uptake were lower in patients undergoing surgery for colorectal cancer compared to healthy individuals [17]. The reduced mobility in the spine joints may also have been caused by the reduced physical activity of patients after the procedure, pain within the operated area and fatigue associated with treatment [18,19].

One factor influencing the described results is certainly the extent of those procedures. In all of them, the following structures weretransected: skin, subcutaneous tissue, fascia and rectus abdominis and oblique muscles, as well as other pelvic structures. The role of scars in limiting the mobility of spine joints adjacent to the operated areas is also important [20]. Collagen fibers bind together all tissues in the operated area. This leads to the formation of adhesions. Initially, it is an important healing process. However, later on, the adhesions can cause dysfunction, especially when they stick to nearby organs and tissues within the abdominal and pelvic cavities. After surgery, the pain may result in unwillingness to take an upright posture, which further promotes adhesion formation and may lead to shortening of the anterior myofascial meridian, and, consequently, cause abnormal body posture, which may result in musculoskeletal pain. An important element will also be the possibility of inhibiting the deep muscles, which are important for maintaining the correct body posture and stabilizing the spine. The lack of sufficiently strong deep muscles can cause back pain, diastasis recti (abdominal separation) and weaken the pelvic floor muscles. Adhesions can also cause defecation problems, urinary incontinence, organ prolapse, hernias and digestive problems [21,22].

Another problem in patients undergoing surgery for colorectal cancer, which limits the range of spine movements and reduces muscle strength, can be the metabolic stress that accompanies surgery. Surgery is a heavy burden for the body. The hormonal and hemodynamic response follows shortly after surgery. Patients lose blood and are immobilized in beds, and there are metabolic changes related to healing (insulin resistance and protein catabolism). Postoperative fatigue and limited physical activity may last for months after surgery [23].

Research shows that, following treatment for colorectal cancer, the patients also show a weakened body core strength [21].

In our study, spine mobility was assessed in patients undergoing open anterior resection and laparoscopic resection. Earlier studies confirmed a number of benefits for patients undergoing laparoscopy, including less postoperative pain, earlier recovery of bowel function, fewer complications after surgery and better quality of life [22,23,24]. In order to improve the results, patients undergoing anterior resection received combined treatment.

Patients undergoing abdominoperineal resection had statistically significantly lower results in spine mobility, abdominal muscle strength and chest mobility compared to other groups, which was caused by the greatest extent of the procedure. Six months after surgery in this group of patients, there was a further reduction in spine mobility for the following measurements: total spine flexion, thoracic spine flexion and lumbar spine flexion.

The reasons for this restriction of mobility can be explained by the fact that the levator ani muscle wasremoved during the abdominoperineal resection of the rectum. This muscle plays an important role in stabilizing the sacroiliac joint, and thus the removal of the levator ani muscle causes disturbances in gait function and discomfort in the anus and sacrum, which are symptoms that develop after rectal resection. Levator ani syndrome causes pain in the sacral area, and the patient’s mobility is limited [2,25]. Another factor contributing to a significant limitation inspine joint mobility and a reduction in muscle strength may be postoperative scars, approx. 15–20 cm long, and the formation of an ostomy [26].

In the study group, patients were not pre-rehabilitated in the preoperative period. The results of the study show that patients who received postoperative rehabilitation in the postoperative period recovered faster after surgery. The implementation of a pre-rehabilitation program also increases the cardiovascular and respiratory capacity and improves muscle strength after surgery [27,28].

Patients undergoing surgery for colorectal cancer are mostly elderly people, who have lower cardiovascular and respiratory capacity, are more prone to postoperative complications and have a slower recovery rate [29]. The mean age of patients in our study was 63.71, and there were no statisticaldifferences in terms of age between the studied groups.

Research has shownthat appropriate conditions for a patient before surgery play a key role in the process of recovery after surgery. Hence, poor physical condition before surgery is associated with an increased postoperative complication rate [30]. In the postsurgical recovery of patients undergoing surgery for colorectal cancer, postoperative rehabilitation plays a key role. Randomized studies and meta-analyses have shown that patients who receive rehabilitation after surgery recover faster [9,31,32].

Despite the fact that our study assessed spine joint mobility, abdominal muscle strength and chest mobility in the long term, it has limitations. It was a single-center study conducted on a small number of patients. It was conducted during the pandemic; therefore, some participants exercised their right to withdraw from the study. Study patients had not been randomized to undergo laparoscopic vs. open resection procedures. The groups were not equal in size, but they consisted of consecutive surgical patients subjected to the same treatment qualification process.

Our study showed no statistical differences for theresults of total spine flexion, inhalation–exhalation difference and rectus abdominis muscle strength in terms of neoadjuvant or adjuvant treatment. Other studies haveshownthe influence of peri- and postopertive treatment onfunctional capacity [33,34], but spinal mobililty, inhalation–exhalation difference and rectus abdominis muscle strength had not been measured yet.

Another limitation of the study was that patients subjected to laparoscopic abdominoperineal resection were not included in the analysis. This was due to the small number of laparoscopic abdominal amputation procedures being performed at the study center. Due to the small total number of patients in the study, variations in spine mobility and muscle strength were not analyzed against the type of neoadjuvant and adjuvant treatment. Significant differences in cancer staging between the groups were another limitation of this study, as a majority of patients subjected to APR had stage I cancer.Moreover, due to the single-center character, the performed surgical procedures had a uniform, standardized character, and it is the surgical technique in patients with rectal cancer that is of key importance for early- and long-term treatment outcomes.

## 5. Conclusions

In our study, we found a significant decrease in spine mobility, abdominal muscle strength and chest mobility for all groups following surgery. Six months after surgery, there was a tendency for improvement of spine mobility, increased muscle strength and improved chest mobility in the AR and LAR groups. Greater improvement was noted in patients undergoing laparoscopic surgery (LAR).

## Figures and Tables

**Figure 1 cancers-14-04148-f001:**
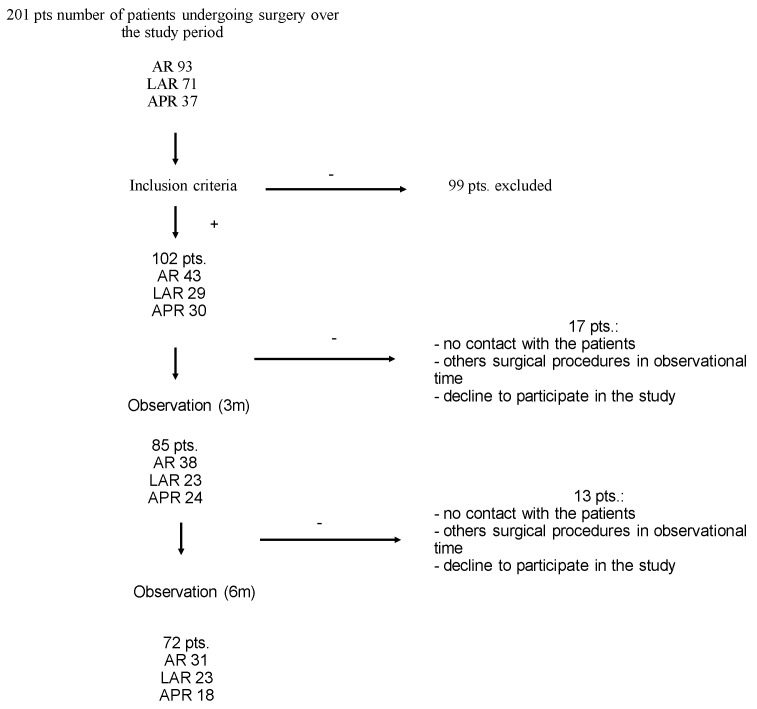
Study flow chart.APR: Abdominoperineal resection; LAR: Laparoscopic anterior resection; AR: Anterior resection.

**Table 1 cancers-14-04148-t001:** Quantitative clinical data in the studied groups of patients (abdominoperineal resection, anterior resection and laparoscopic anterior resection) and the relationships between those groups.

Parameter	APR		LAR		AR		ANOVA
	Mean	S.D.	Mean	S.D.	Mean	S.D.	
Age	63.22	8.29	63.52	11.53	64.39	7.00	*p* = 0.8920
Weight	79.06	14.06	76.06	14.61	80.17	17.48	*p* = 0.6343
Height	171.9	8.36	168.4	10.56	169.5	11.28	*p* = 0.5646
BMI	26.93	5.90	26.68	3.88	27.80	4.90	*p* = 0.4999
Hospitalizationtime **	8.44	4.25	7.48	4.33	9.74	6.79	*p* = 0.0922

APR: Abdominoperineal resection; LAR: Laparoscopic anterior resection; AR: Anterior resection; *p*: *p*-value. ** for ‘hospitalization time’, the analysis included a Kruskal–Wallis test and a posthoc Dunn test with Bonferroni correction.

**Table 2 cancers-14-04148-t002:** Qualitative clinical data in the studied groups of patients (abdominoperineal resection, anterior resection and laparoscopic anterior resection) and the relationships between those groups.

		Type of Surgical Procedure			^1^ Chi-Squared Test / ^2^ Fisher’s Exact Test
		APR	AR	LAR	
		%	%	%	
Sex	Female	22.2	35.5	39.1	^1^*p* = 0.4935
	Male	77.8	64.5	60.9	
Type of preoperativetreatment	none	5.6	35.5	39.1	^1^*p* = 0.0439
	radiochemotherapy	61.1	48.4	26.1	
	radiotherapy	33.3	16.1	34.8	
Type of postoperativetreatment	none	77.8	64.5	68.2	^1^*p* = 0.5161
	chemotherapy	22.2	35.5	27.3	
	radiochemotherapy	0	0	4.5	
Postoperativecomplications	none	66.7	71	87	^1^*p* = 0.2607
	present	33.3	29	13	
Cancerstaging pTNM/ypTNM	0	0	9.7	4.3	^2^*p* = 0.0466
	I	52.9	16.1	39.1	
	IIA	17.6	38.7	17.4	
	IIB	5.9	0	0	
	IIIA	5.9	0	0	
	IIIB	17.6	16.1	26.1	
	IIIC	0	19.3	13.0	

APR: Abdominoperineal resection; LAR: Laparoscopic anterior resection; AR: Anterior resection; %: percentages; *p*: *p*-value.

**Table 3 cancers-14-04148-t003:** Mobility of spine joints, evaluation of strength of the abdominal muscles and assessment of chest mobility in the study groups (abdominoperineal resection, anterior resection and laparoscopic anterior resection) and the differences between those groups foreach examination.

	Term	Type of Surgical Procedure			Kruskal-Wallis Test
		APR	LAR	AR	
		Median	Median	Median	
Total spineflexion	I	7.0 a/b	7.0 a/b	7.0 a/b	*p* = 0.9715
	II	2.0 a/a	5.0 b/a	4.0 b/a	*p* = 0.0001
	III	2.0 a/a	6.0 b/b	5.0 b/a	*p* < 0.0001
Friedman test		*p* < 0.0001	*p* < 0.0001	*p* < 0.0001	
Thoracicspineflexion	I	1.0 a/b	1.0 a/a	2.0 a/b	*p* = 0.6448
	II	0.0 a/a	1.0 b/a	1.0 b/a	*p* = 0.0003
	III	1.0 a/a	1.0 b/a	1.0 b/a	*p* = 0.0052
Friedman test		*p* = 0.0002	*p* = 0.1282	*p* = 0.0001	
Lumbarspineflexion	I	3.0 a/b	4.0 a/b	4.0 a/b	*p* = 0.1769
	II	2.0 a/a	3.0 b/a	2.0 ab/a	*p* = 0.0038
	III	1.0 a/a	3.0 b/a	2.0 b/a	*p* = 0.0001
Friedman test		*p* < 0.0001	*p* = 0.0001	*p* < 0.0001	
Lumbarspineextension	I	2.0 a/b	3.0 a/b	2.0 a/b	*p* = 0.6542
	II	1.0 a/a	2.0 b/a	1.0 b/a	*p* = 0.0076
	III	1.0 a/a	2.0 b/ab	2.0 b/a	*p* = 0.0001
Friedman test		*p* < 0.0001	*p* = 0.0001	*p* < 0.0001	
Lateral thoracolumbar spine flexion, R	I	4.0 a/b	5.0 a/b	4.0 a/b	*p* = 0.5300
	II	2.0 a/a	4.0 b/a	3.0 a/a	*p* = 0.0009
	III	2.0 a/a	5.0 b/b	3.0 a/a	*p* = 0.0012
Friedman test		*p* < 0.0001	*p* < 0.0001	*p* < 0.0001	
Lateral thoracolumbar spine flexion, L	I	4.5 a/b	5.0 a/b	4.0 a/b	*p* = 0.5794
	II	2.0 a/a	4.0 b/a	3.0 ab/a	*p* = 0.0299
	III	2.5 a/a	5.0 b/b	3.0 ab/a	*p* = 0.0153
Friedman test		*p* < 0.0001	*p* < 0.0001	*p* < 0.0001	
Thoracolumbarspinerotation, R	I	3.0 a/b	3.0 a/b	3.0 a/b	*p* = 0.5496
	II	1.0 a/a	2.0 b/a	1.0 ab/a	*p* = 0.0022
	III	1.0 a/a	3.0 b/ab	2.0 b/a	*p* = 0.0001
Friedman test		*p* < 0.0001	*p* < 0.0001	*p* < 0.0001	
Thoracolumbarspinerotation, L	I	3.0 a/b	3.0 a/b	3.0 a/b	*p* = 0.9977
	II	1.0 a/a	2.0 b/a	1.0 ab/a	*p* = 0.0191
	III	1.0 a/a	2.0 b/ab	1.0 b/a	*p* = 0.0001
Friedman test		*p* < 0.0001	*p* < 0.0001	*p* < 0.0001	
Inhalation–exhalationdifference	I	3.0 a/b	4.0 a/b	4.0 a/b	*p* = 0.8596
	II	3.0 a/a	3.0 a/a	3.0 a/a	*p* = 0.4900
	III	3.0 a/ab	3.0 a/a	2.0 a/a	*p* = 0.0803
Friedman test		*p* = 0.0002	*p* < 0.0001	*p* < 0.0001	
Rectusabdominismuscle strength	I	23.5 a/b	22.2 a/c	23.1 a/c	*p* = 0.2104
	II	8.2 a/a	14.3 b/a	11.4 a/a	*p* = 0.0007
	III	9.4 a/a	18.9 b/b	16.5 b/b	*p* < 0.0001
Friedman test		*p* < 0.0001	*p* < 0.0001	*p* < 0.0001	
Abdominal oblique muscle strength, R	I	22.5 a/c	21.3 a/c	21.2 a/c	*p* = 0.6302
	II	9.5 a/a	13.7 b/a	8.9 a/a	*p* < 0.0001
	III	11.3 a/b	18.9 b/b	13.4 a/b	*p* = 0.0015
Friedman test		*p* < 0.0001	*p* < 0.0001	*p* < 0.0001	
Abdominal oblique muscle strength, L	I	22.1 a/c	22.2 a/c	21.6 a/c	*p* = 0.9001
	II	3.5 a/a	13.5 b/a	12.9 b/a	*p* < 0.0001
	III	7.0 a/b	20.8 c/b	16.6 b/b	*p* < 0.0001
Friedman test		*p* < 0.0001	*p* < 0.0001	*p* < 0.0001	

APR: Abdominoperineal resection; LAR: Laparoscopic anterior resection; AR: Anterior resection; I: Before surgery; II: 3 months after surgery; III: 6 months after surgery; L: Left side; R: Right side; X: Arithmetic mean; *p*: Statistical significance level indicator. a,b—below the median values, there are letter codes indicating homogeneous groups based on the Dunn–Bonferroni test. The first line below the median represents the results of the Kruskal–Wallis test, and the second line below the median shows the results of the Friedman test.

**Table 4 cancers-14-04148-t004:** Evaluation of total spine flexion, rectus abdominis muscle strength assessment and chest mobility assessment in the study groups in terms of neoadjuvant and adjuvant treatment.

	Term	Treatment				Kruskal–Wallis Test
		Not Neoadjuvant or Adjuvant Treatment	Neoadjuvant Treatment	Adjuvant Treatment	Neoadjuvant or Adjuvant Treatment	
		Median	Median	Median	Median	
Total spine flexion	I	7 a/b	7 a/b	6 a/b	7 a/b	*p* = 0.6778
	II	4.5 a/a	3 a/a	4 a/a	4 a/a	*p* = 0.6093
	III	5 a/a	5 a/a	4 a/ab	5 a/a	*p* = 0.8836
Friedman’s test		*p* = 0.0001	*p* < 0.0001	*p* = 0.0080	*p* = 0.0001	
Inhalation–exhalation difference	I	5 a/b	3 a/b	4 a/b	4 a/b	*p* = 0.2243
	II	3.5 a/a	3 a/a	3 a/ab	3 a/a	*p* = 0.2744
	III	3 a/a	2 a/a	3 a/a	3 a/a	*p* = 0.3355
Friedman’s test		*p* = 0.0014	*p* < 0.0001	*p* = 0.0038	*p* = 0.0004	
Rectus abdominis muscle strength	I	23.7 a/b	22.35 a/c	22.2 a/b	22.6 a/b	*p* = 0.7330
	II	12.3 a/a	11.2 a/a	12.4 a/a	12.4 a/b	*p* = 0.6711
	III	18.1 a/a	15.45 a/b	16.4 a/a	14.2 a/b	*p* = 0.4113
Friedman’s test		*p* < 0.0001	*p* < 0.0001	*p* = 0.0038	*p* < 0.0001	

I: Before surgery; II: 3 months after surgery; III: 6 months after surgery; X: Arithmetic mean; *p*: statistical significance level indicator; a,b,c—below the median values, there are letter codes indicating homogeneous groups based on the Dunn–Bonferroni test. The first line below the median represents the results of the Kruskal–Wallis test, and the second line below the median shows the results of the Friedman test. The same letter at two median values indicates no significant difference (*p* > 0.05); if no same letters are used with two median values, the difference between the compared medians (distributions) is statistically significant (*p* < 0.05).

## Data Availability

The datasets generated during and/or analyzedduring the current study are available from the corresponding author on responsible request.

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
