# Peer review of "The Impact of Surgical Techniques in Patients with Rectal Cancer on Spine Mobility and Abdominal Muscle Strength—A Prospective Study"

_cancers, 2022, doi:10.3390/cancers14174148_

Round 1

Reviewer 1 Report

1、The abstract should be a single paragraph and should follow the style of structured abstracts, but without headings.

2、Why longer time is necessary to report the spine mobility and muscle strength? This should be illuminated further in introduction, especially regarding its clinical value.

3、For better accuracy, spine mobility is usually measured in X-ray images.

4、The tables must be improved. Extremely hard to read. Missing information in Table 2 (Type of postoperative treatment). Decimal point, instead of comma, should be used to separate the whole number and the decimal fraction. Very large front was used as compared with that in the main text. Unnecessary abbreviations were used. There were no units for every figure.

5、There are two Table 1.

6、The numbers in Table 1 (Study flow chart) don’t add up.

7、what does abc stand for in Table 3?

8、For statistical analysis, normality test should be performed for continuous variables. Additionally, parametric test should be considered first, before non-parametric test.

9、Neoadjuvant and postoperative radiochemotherapy may have an impact on Spine Mobility and Abdominal Muscle Strength. Related data should be reported and sufficiently discussed to alleviate potential bias.

10、          The discussion section was poorly written, very disorganized. For instance, in Line 333, the benefit of radiochemotherapy was first mentioned to reduce local recurrence. However, immediately after that, they started to talk about spine mobility, which is unrelated to the previous topic. Additionally, benefit of radiochemotherapy on local control had almost nothing to do with the focus of this study. Moreover, a lot of study results were repeated again and again in the discussion.

11、          The study results were also repeated in the conclusion section, which should be briefly summarized.

Author Response

Dear Reviewer #1,

Thank you for reviewing our article titled ‘The Impact of Surgical Techniques in Patients with Rectal Cancer on Spine Mobility and Abdominal Muscle Strength – A Prospective Study’ We deeply appreciate your opinion as well as constructive comments that contributed to more profound consideration of issues addressed in our publication. The comments in your review will guide us in our future work.

            In response to those commentaries we have clarified the following points.

  1. The abstract should be a single paragraph and should follow the style of structured abstracts, but without headings.

We have corrected this part

2、Why longer time is necessary to report the spine mobility and muscle strength? This should be illuminated further introduction, especially regarding its clinical value.

We have clarified this part in the introduction section.

3、For better accuracy, spine mobility is usually measured in X-ray images.

Yes, we agree with your opinion. However, X-ray imaging is not a standard procedure in patients with colorectal cancer. We didn't want to expose oncological patients to additional procedures not required in the therapeutic management of cancer.

4、The tables must be improved. Extremely hard to read. Missing information in Table 2 (Type of postoperative treatment). Decimal point, instead of comma, should be used to separate the whole number and the decimal fraction. Very large front was used as compared with that in the main text. Unnecessary abbreviations were used. There were no units for every figure.

We have improved all the tables, we added information – type of postoperative treatment in Table 2. We have explained all the abbreviations.

5、There are two Table 1.

We have corrected it. Instead of Table 1 we have created Fig 1.

6、The numbers in Table 1 (Study flow chart) don’t add up.

We have corrected this part.

7、what does abc stand for in Table 3?

Below the median values, there are letter codes indicating homogeneous groups based on Dunn-Bonferroni test. The first line below the median represents the results of Kruskal-Wallis test, and the second line below the median shows the results of Friedman's test.

The same letter at two median values indicates no significant difference (p > 0.05); if no same letters are used with two median values, the difference between the compared medians (distributions) is statistically significant (p < 0.05).

8、For statistical analysis, normality test should be performed for continuous variables. Additionally, parametric test should be considered first, before non-parametric test.

A parametric test was considered first but due to the lack of normal distribution we performed a non-parametric test.

9、Neoadjuvant and postoperative radiochemotherapy may have an impact on Spine Mobility and Abdominal Muscle Strength. Related data should be reported and sufficiently discussed to alleviate potential bias.

We have discussed this part in discussion section. We have considered holistic treatment (including surgery and neoadiuvant and adiuvant treatment) for selected groups of patients.

10、          The discussion section was poorly written, very disorganized. For instance, in Line 333, the benefit of radiochemotherapy was first mentioned to reduce local recurrence. However, immediately after that, they started to talk about spine mobility, which is unrelated to the previous topic. Additionally, benefit of radiochemotherapy on local control had almost nothing to do with the focus of this study. Moreover, a lot of study results were repeated again and again in the discussion.

We have corrected discussion section.

11、          The study results were also repeated in the conclusion section, which should be briefly summarized.

We have corrected conclusion section.

Thanks to the comments received, we were able to refine the publication in terms of the content. All errors mentioned in the review were corrected in the final version of the publication.

                                                                                                          Kind regards,

                                                                                                          the Authors

Reviewer 2 Report

In the presented manuscript ‘The impact of surgical techniques in patients with rectal cancer on spine mobility and abdominal muscle strength – a prospective study’, Glowacka-Mrotek et. al reported on changes in spine mobility, abdominal muscle strength and chest mobility of 72 colorectal cancer patients undergoing undergoing anterior resection, abdominoperineal resection or laparoscopic anterior resection. Data shows dynamics of postoperative improvement after six month which was greatest in patients undergoing laparoscopic anterior resection. Overall, this study sheds a light on the importance of postoperative multidisciplinary care which includes physical rehabilitation programs aimed at improving strength and mobility. 

Here are some minor comments:

1.     As this study postulates an important role of rehabilitation, please include an overview of your clinics rehabilitation program (i.g. physiotherapists on wards?, fast track mobilization?, home programs?).

2.     The data would profit greatly if patient reported data could be included regarding abilities of daily living and quality of life as mentioned in discussion: ‘The reduced mobility in the spine joints may also have been caused by reduced physical activity of patients after the procedure, pain within the operated area, and the fatigue associated with treatment.’

3.     Also, postoperative pain levels (NRS) and (length of additional) opiate use would be interesting additional data.

4.     Please report which postoperative complications occurred (i.g. revision rate and infection).

5.     Please update Table 1 to show data as a flow chart (top to bottom; not as table) to improve understanding.

6.     Table 2 and 3 would profit from a layout revision to improve readability. It might help to remove not significant data from table 3 (attach to appendix) and please explain lettering a, b and c. 

7.     As measurements are explained in detail, there is no need to include further description: total spine flexion (forward bend) -> total spine flexion should be enough.

8.     Please shorten as to list right AND left (i.g.) abdominal oblique muscle strength, and left abdominal oblique muscle strength. This shortens sentences as keeping track of the data might otherwise be confusing.

Author Response

Dear Reviewer #2,

Thank you for reviewing our article titled ‘The Impact of Surgical Techniques in Patients with Rectal Cancer on Spine Mobility and Abdominal Muscle Strength – A Prospective Study’ We deeply appreciate your opinion as well as constructive comments that contributed to more profound consideration of issues addressed in our publication. The comments in your review will guide us in our future work.

Here are some minor comments:

  1. As this study postulates an important role of rehabilitation, please include an overview of your clinics rehabilitation program (i.g. physiotherapists on wards?, fast track mobilization?, home programs?).

We have described our rehabilitation program in Material and Method section.

.

  1. The data would profit greatly if patient reported data could be included regarding abilities of daily living and quality of life as mentioned in discussion: ‘The reduced mobility in the spine joints may also have been caused by reduced physical activity of patients after the procedure, pain within the operated area, and the fatigue associated with treatment.’

We agree with your opinion. Unfortunately we didn’t collect any specific info from patients about this. But we would like to add that we have realized the prehabilitation program for patients with colorectal cancer and we have collected the info about daily living of patient and their quality of life.

  1. Also, postoperative pain levels (NRS) and (length of additional) opiate use would be interesting additional data.

We did our second examination tree months after surgery. In our previous study where we examined patients 5-6 days after surgey we used VAS scale to analyze postoperative pain  level. Her is the link to our previous article: https://www.termedia.pl/Journal/-42/pdf-3649610?filename=Prospective%20evaluation.pdf

  1. Please report which postoperative complications occurred (i.g. revision rate and infection).

We have clarified this part in results section. Postoperative complications as observed in the study in patients subjected to abdominoperineal resection included surgical site infection (3 cases), postoperative obstruction (2 cases), and stoma necrosis (1 case). In patients subjected to anterior laparoscopic resection of the rectum, intestinal anastomosis leak was reported in 3 cases. In patients subjected to open anterior resection of the rectum, surgical site infection was reported in 4 cases, intestinal anastomosis leak was reported in 3 cases, drain displacement was reported in 1 case, and an abscess within the pelvis was reported in 1 case.

  1. Please update Table 1 to show data as a flow chart (top to bottom; not as table) to improve understanding.

We have updated this table as a flow chart.

  1. Table 2 and 3 would profit from a layout revision to improve readability. It might help to remove not significant data from table 3 (attach to appendix) and please explain lettering a, b and c. 

We have correct the visibility and redability of all tables, instead of table 1 we have created study flow chart. We have explained lettering a, b and c. We corrected table 3 and clarified all the abbreviation. Below the median values, there are letter codes indicating homogeneous groups based on Dunn-Bonferroni test. The first line below the median represents the results of Kruskal-Wallis test, and the second line below the median shows the results of Friedman's test.The same letter at two median values indicates no significant difference (p > 0.05); if no same letters are used with two median values, the difference between the compared medians (distributions) is statistically significant (p < 0.05).

  1. As measurements are explained in detail, there is no need to include further description: total spine flexion (forward bend) -> total spine flexion should be enough.

We have corrected this part in Materials and Methods and method and results section.

  1. Please shorten as to list right AND left (i.g.) abdominal oblique muscle strength, and left abdominal oblique muscle strength. This shortens sentences as keeping track of the data might otherwise be confusing.

We have shortened this words in the text.

            Thanks to the comments received, we were able to refine the publication in terms of the content. All errors mentioned in the review were corrected in the final version of the publication.

                                                                                                          Kind regards,

                                                                                                          the Authors

Reviewer 3 Report

Congratulations for the approach in analysis of impact of CRC surgery, yet another layer. That said, it would be great to have a paragraphing introduction explain the correlation of mobility to possible interventions later on or why did you choose to do this?

In the materials and methods section, all is described but it would be easier to have more content clearly presented (inclusion/exclusion criteria diagram or alike).

Similar note to results section, tables are pretty unclear, and minor editing might improve both data presentation and visualisation of content.

Finally, related to introduction paragraph, what can we use this quantification for, do you have a suggestion with regards to stratified physiotherapy either in form of prehabilitation or rehabilitation.

Looking forward to your answers, once again interesting approach.

Author Response

Dear Reviewer #3,

Thank you for reviewing our article titled ‘The Impact of Surgical Techniques in Patients with Rectal Cancer on Spine Mobility and Abdominal Muscle Strength – A Prospective Study’ We deeply appreciate your opinion as well as constructive comments that contributed to more profound consideration of issues addressed in our publication. The comments in your review will guide us in our future work.

            In response to those commentaries we clarified (please see below your commentaries and our response)

  1. In the materials and methods section, all is described but it would be easier to have more content clearly presented (inclusion/exclusion criteria diagram or alike).

We have improved the inclusion/exclusion criteria section redability.

  1. Similar note to results section, tables are pretty unclear, and minor editing might improve both data presentation and visualisation of content.

We have corrected the visibility and readability of all tables, instead of table 1 we have created study flow chart. We have explained lettering a, b and c. We corrected table 3 and clarified all the abbreviation

  1. Finally, related to introduction paragraph, what can we use this quantification for, do you have a suggestion with regards to stratified physiotherapy either in form of prehabilitation or rehabilitation.

We have clarified this part in introduction section.

Thanks to the comments received, we were able to refine the publication in terms of the content. All errors mentioned in the review were corrected in the final version of the publication.

Reviewer 4 Report

The study shows that spine joint mobility and muscle strength after surgery for rectal cancer are diminished three and six months after operation. In addition, analysed parameters are worst in patients after abdominoperineal resection. The analysis of joint mobility and muscle strength is thoroughly analysed and presented in tables with all necessary details. The main disadvantage of the study is low number of patients. Therefore the power of the statistical tests is probably low.

There are some issues that should be addressed before possible publication:

1. Simple summary (lines 26-28) and discussion (lines 393-395): the authors  stated that the study showed necessity for rehabilitation in the pre- and post- operative period. In fact they did not show that rehabilitation improves spine joint motility or muscle strength. These sentences should be rephrased   as the study did not analysed the influence of rehabilitation on joint or muscle function. 

2. Abstract: line 38 - there is information that groups did not differ in terms of age, height, BMI but p value is <0.05.

3. Introduction - line 67, in square brackets there are two letters "aa" - probably citation is missing.

4. Introduction - line 69-70 - the authors stated that during rectal cancer surgery rectus muscles are transected but i.e. in open surgery midline incision can be performed and only fascia is incised not the muscle itself. 

5. Line 105 - ASA was determined for all patients but data are not shown in tables, it is essential to include these data in patients characteristics as patients with more comorbidities (ASA III) may have problems with postoperative rehabilitation and also limited spine joint mobility. 

6. Line 107 and table 2 - postoperative complications were analysed - what classification was used? Clavien-Dindo? Are the complications in table 2 of all categories (I-V)?

7. Lines 107-108: "staging of cancer" is written twice

8. Line 125 -patients with weight loss >5% over past 3 months were exclude from the analysis. What was the reason for that. It may lead to selection bias as many patients with rectal cancer have significant weight loss.

9. Lines 193-194: The authors should specify what test was used for checking normal distribution of variables: age, weight, height, BMI – if the distribution is not normal (as usually in such alayzes) it is not possible to use ANOVA for comparison of the groups. To compare length of hospitalization Kruskall-Wallis test was conducted – why not ANOVA in this case?

10. Table 2 – there were significant differences between groups in terms of staging, majority of patients who underwent APR had stage I cancer, this should be listed as limitation of the study in the Discussion. 

11. Lines 140-144: Abdominoperineal  resection (perineal part) may be performed in Lloyd-Davies position or jackknife position – the position on the table may have potential influence on postoperative spine joint mobility. The authors describe the procedure without this important information.

12. Lines 285-288: The authors describe contemporary neoadjuvant treatment of rectal cancer consisting of radio(chemo)therapy. What about total neoadjuvant treatment that is currently included in current NCCN guidelines? Were patients in the study treated according to this conception? May prolonged preoperative therapy (as in RAPIDO trial) may influence muscle strength?

13. The authors should specify whether ERAS protocol was implemented in analysed cohort? Maybe only some components of ERAS were implemented? Please specify.

Author Response

Dear Reviewer #3,

Thank you for reviewing our article titled ‘The Impact of Surgical Techniques in Patients with Rectal Cancer on Spine Mobility and Abdominal Muscle Strength – A Prospective Study’ We deeply appreciate your opinion as well as constructive comments that contributed to more profound consideration of issues addressed in our publication. The comments in your review will guide us in our future work.

                In response to those commentaries we clarified (please see below your commentaries and our response)

The main disadvantage of the study is low number of patients. Therefore the power of the statistical tests is probably low.

We have done our research during pandemic time and the inclusion, exclusion criteria were very strict.

There are some issues that should be addressed before possible publication:

  1. Simple summary (lines 26-28) and discussion (lines 393-395): the authors  stated that the study showed necessity for rehabilitation in the pre- and post- operative period. In fact they did not show that rehabilitation improves spine joint motility or muscle strength. These sentences should be rephrased   as the study did not analysed the influence of rehabilitation on joint or muscle function. 

We have corrected this part.

  1. Abstract: line 38 - there is information that groups did not differ in terms of age, height, BMI but p value is <0.05.

We have corrected this part.

  1. Introduction - line 67, in square brackets there are two letters "aa" - probably citation is missing.

We have added the citation.

  1. Introduction - line 69-70 - the authors stated that during rectal cancer surgery rectus muscles are transected but i.e. in open surgery midline incision can be performed and only fascia is incised not the muscle itself. 

When a midline incision is performed below the umbilicus, the rectus muscle sheath is frequently transected, sometimes on both sides. This is due to the fact that it is difficult to perform the incision directly between the sheaths of the rectus muscle in an anesthetized patient with muscle relaxants being used in the linea alba region. In addition, mini-laparotomy in the laparoscopic approach is performed with the continuity of the external and internal oblique muscles as well as the transverse muscle being interrupted.

  1. Line 105 - ASA was determined for all patients but data are not shown in tables, it is essential to include these data in patients characteristics as patients with more comorbidities (ASA III) may have problems with postoperative rehabilitation and also limited spine joint mobility. 

We have included in our study patient with ASA (I-III). To avoid this situation we measured all the patients at the beginning, before surgery and we compared their results 3 months and 6 months after surgery.

  1. Line 107 and table 2 - postoperative complications were analysed - what classification was used? Clavien-Dindo? Are the complications in table 2 of all categories (I-V)?

Complications were classified according to a modified Clavien-Dindo scale. Grade II–IV complications included 3× surgical site infections, 2× postoperative obstruction, and 1× stoma necrosis in the APR group, 3× intestinal anastomosis leak in the LAR group, 4× surgical site infection, 3: intestinal anastomosis leak, 1× drain displacement and 1× pelvic abscess in the AR group.

  1. Lines 107-108: "staging of cancer" is written twice

We have corrected this part.

  1. Line 125 -patients with weight loss >5% over past 3 months were exclude from the analysis. What was the reason for that. It may lead to selection bias as many patients with rectal cancer have significant weight loss.

Unintended weight loss in patients with stage I–III rectal cancer is uncommon. If it occurs, it may be associated with changes in body composition, complications from neoadjuvant therapy (radiotherapy), or cancer spread. Weight loss may affect the general performance, the condition and function of muscles, the strength of the integumental muscles, and the mobility of the spine. We considered the weight loss of more than 5% within the last 3 months to be an exclusion criterion for better standardization of the results, as well as due to the fact that this parameter is included within the NRS 2002, nutrition risk screening survey which normally completed for all patients upon admission.

  1. Lines 193-194: The authors should specify what test was used for checking normal distribution of variables: age, weight, height, BMI – if the distribution is not normal (as usually in such alayzes) it is not possible to use ANOVA for comparison of the groups. To compare length of hospitalization Kruskall-Wallis test was conducted – why not ANOVA in this case?

For the time of  hospitalization, we have a non-parametric test because the distributions of the results are "non-normal "And for the other scales in Table 1, we have a parametric test because the distributions are "normal." The test that checked normality is the "Kolmogorov-Smirnov test." 

  1. Table 2 – there were significant differences between groups in terms of staging, majority of patients who underwent APR had stage I cancer, this should be listed as limitation of the study in the Discussion. 

We have listed this as a limitation of the study.

  1. Lines 140-144: Abdominoperineal  resection (perineal part) may be performed in Lloyd-Davies position or jackknife position – the position on the table may have potential influence on postoperative spine joint mobility. The authors describe the procedure without this important information.

In all patients with APR, the perineal part was performed in the gynecological (lithotomy) position whereas the abdominal part was performed in the supine position. Jackknife position was not used.

  1. Lines 285-288: The authors describe contemporary neoadjuvant treatment of rectal cancer consisting of radio(chemo)therapy. What about total neoadjuvant treatment that is currently included in current NCCN guidelines? Were patients in the study treated according to this conception? May prolonged preoperative therapy (as in RAPIDO trial) may influence muscle strength?

Preoperative treatment did not involve the TNT as defined by current recommendations (NCCN). 2 regimens were used: short-course RT followed by ctx (5 FU or FOLFOX) and long-course chemo/RT (5-FU or capecitabine). Based on Erlandsson (2017) [a], we followed a regimen including surgery being delayed by 12 weeks after initiation of RT (short-course or long-course), with restaging (DRE, MRI, endoscopy if required) 11 weeks after initiation of RT. If clinical complete response occurred, an observational approach based on CEA, DRE, endoscopy (3–4 times per year in the first 3 years, 2 times per year thereafter), pelvic MRI (2 times per year), as well as abdominal and thoracic CT (1–2 times per year), was preferred. In our material, all patients were subjected to the same regimen with delayed surgery; we believe this standardized the results.

  1. The authors should specify whether ERAS protocol was implemented in analysed cohort? Maybe only some components of ERAS were implemented? Please specify.

 The ERAS protocol was not applied in a restrictive, exhaustive way. Selected elements of the ERAS protocol were used as follows:

Sustained physical activity and protein/energy supplementation were recommended prior to the surgery along with carbohydrate drink up to 2 h before anesthesia, avoidance of preoperative fasting, restrictive fluid therapy, prompt restitution of oral nutrition, shorter periods of abdominal drain, and intravesical catheter use, avoidance of gastric probes being used as a standard measure, and timely discharge.

Thanks to the comments received, we were able to refine the publication in terms of the content. All errors mentioned in the review were corrected in the final version of the publication.

Kind regards,

                                                                                                                                             the Authors

Reviewer 5 Report

This is a prospective study on a known topic and its conclusions once again corroborate the superiority of laparoscopy in the recovery of patients.

The introduction is correct but data on the ERAS programs is missing.

In material and methods, the disproportion of the study groups is striking: 18-23-31 and especially in open and laparoscopic surgery 49-23.

Table 1 should be turned into a graph and the figures for the RA group are not correct: 93 patients operated - 41 patients who decline to participate or do not meet the criteria are 52 and not 31?.

In Table 1, the lost patients (at the first and second appointments) should be placed after those excluded because they did not meet the criteria or who declined to participate.

It is surprising that there are no patients operated on by laparoscopy in Miles' abdominoperineal amputation group.

Patients requiring conversion should be included if the study is intention-to-treat. If not, it should be specified that it is a per-protocol analysis.

A study on the sample size has not been carried out. In addition, the results do not present confidence intervals and are only expressed with the p value.

Table 3 is too big. Perhaps in graph form it could be more understandable.

The conclusion about worse data with the Miles procedure cannot be sustained in the absence of a laparoscopic control group.

Regarding the limitations of the study, it is commented that it is not a randomized study? Refers to open and laparoscopic surgery. It should be clearly expressed.

Author Response

Dear Reviewer #5,

Thank you for reviewing our article titled ‘The Impact of Surgical Techniques in Patients with Rectal Cancer on Spine Mobility and Abdominal Muscle Strength – A Prospective Study’ We deeply appreciate your opinion as well as constructive comments that contributed to more profound consideration of issues addressed in our publication. The comments in your review will guide us in our future work.

In response to those commentaries we clarified the following (please see below your commentaries and our response)

  1. The introduction is correct but data on the ERAS programs is missing.

We have clarified the information about ERAS program in Introduction and Materials and methods section. The ERAS protocol was not applied in a restrictive, exhaustive way. Selected elements of the ERAS protocol were used as follows:

Sustained physical activity and protein/energy supplementation were recommended prior to the surgery along with carbohydrate drink up to 2 h before anesthesia, avoidance of preoperative fasting, restrictive fluid therapy, prompt restitution of oral nutrition, shorter periods of abdominal drain, and intravesical catheter use, avoidance of gastric probes being used as a standard measure, and timely discharge.

  1. In material and methods, the disproportion of the study groups is striking: 18-23-31 and especially in open and laparoscopic surgery 49-23.

The recruitment of patients was originally planned to be carried out until May 31, 2020, but due to the coronavirus pandemic, it was terminated earlier .We had no influence on the final size of the groups.

  1. Table 1 should be turned into a graph and the figures for the RA group are not correct: 93 patients operated - 41 patients who decline to participate or do not meet the criteria are 52 and not 31?.

We have turned table 1 into graph and corrected the sizes of study groups.

  1. In Table 1, the lost patients (at the first and second appointments) should be placed after those excluded because they did not meet the criteria or who declined to participate.

We have presented this in graph 1.

  1. It is surprising that there are no patients operated on by laparoscopy in Miles' abdominoperineal amputation group.

Since only about 15 patients are operated on by laparoscopy in Miles' abdominoperineal amputation each year at our study center, they were not included in the study due to the small potential sample size. This was mentioned as a study limitation in the manuscript

  1. Patients requiring conversion should be included if the study is intention-to-treat. If not, it should be specified that it is a per-protocol analysis.

Our study was per-protocol analysis. We have added this info in Materials and Methods section.

  1. A study on the sample size has not been carried out. In addition, the results do not present confidence intervals and are only expressed with the p value.

The recruitment of patients was originally planned to be carried out until May 31, 2020- (during one year) ,  but due to the coronavirus pandemic, it was terminated earlier.

  1. Table 3 is too big. Perhaps in graph form it could be more understandable.

We have improved all the tables, we added information. We have explained all the abbreviations

  1. The conclusion about worse data with the Miles procedure cannot be sustained in the absence of a laparoscopic control group.

The purpose of our study was to compare the results of 3 groups abdominoperineal resection; LAR - laparoscopic anterior resection; AR - anterior resection. Among the three groups, patients treated with abdominoperineal resection had worse results in terms of spinal mobility and muscle strength. The purpose of our study was to highlight that the extensive scar and surgical wound in these patients contributes to poorer results in spinal mobility and reduced muscle strength. We cited the absence of Miles' abdominoperineal amputation group as a limitation of our study.

  1. Regarding the limitations of the study, it is commented that it is not a randomized study? Refers to open and laparoscopic surgery. It should be clearly expressed.

We have cleary expressed this part in limitation of the study part.

Thanks to the comments received, we were able to refine the publication in terms of the content. All errors mentioned in the review were corrected in the final version of the publication.

                                                                                                          Kind regards,

                                                                                                          the Authors

Round 2

Reviewer 1 Report

Agree to accept

Author Response

Dear Reviewer,

Thank you for your comment.

The Authors

Reviewer 4 Report

The authors corrected the manuscript properly. They responded to the majority of criticism raised by the reviewers. In adition, some errors were corrected and clarified. Although the number of included patients is relatively low, I would like to recommend publication of the paper.

Author Response

Dear Reviewer,

Thank you for your comments.

In our future work we will try to include more patients in the study.

The Authors